scikit-image: image processing in Python

van der Walt Stéfan 1 stefan@sun.ac.za
Schönberger Johannes L. 2
Nunez-Iglesias Juan 3
Boulogne François 4
Warner Joshua D. 5
Yager Neil 6
Gouillart Emmanuelle 7
Yu Tony 8
the scikit-image contributors
1 Stellenbosch University , Stellenbosch , South Africa
2 Department of Computer Science, University of North Carolina at Chapel Hill , Chapel Hill, NC , USA
3 Victorian Life Sciences Computation Initiative , Carlton, VIC , Australia
4 Department of Mechanical and Aerospace Engineering, Princeton University , Princeton, NJ , USA
5 Department of Biomedical Engineering, Mayo Clinic , Rochester, MN , USA
6 AICBT Ltd , Oxford , UK
7 Joint Unit, CNRS/Saint-Gobain , Cavaillon , France
8 Enthought, Inc. , Austin, TX , USA
Gomez Shawn
Electronic publication date: 2014 Jun 19
Publication date: 2014
Volume: 2
Electronic Location ID: e453
Received 2014 Apr 2; Accepted 2014 Jun 4
Copyright: © 2014 Van der Walt et al.
Copyright year: 2014
Copyright holder: Van der Walt et al.
License: This is an open access article distributed under the terms of the Creative Commons Attribution License, which permits unrestricted use, distribution, reproduction and adaptation in any medium and for any purpose provided that it is properly attributed. For attribution, the original author(s), title, publication source (PeerJ) and either DOI or URL of the article must be cited.
License URL: https://creativecommons.org/licenses/by/4.0/

Keywords: Image processing, Reproducible research, Education, Visualization, Open source, Python, Scientific programming

Funding: NIH F30DK098832 Portions of the research reported in this publication were supported by the National Institute of Diabetes and Digestive and Kidney Diseases of the National Institutes of Health under award number F30DK098832. Portions of the research reported in this paper were supported by the Victorian Life Sciences Computation Initiative. The funders had no role in study design, data collection and analysis, decision to publish, or preparation of the manuscript.

==============================
scikit-image is an image processing library that implements algorithms and utilities for use in research, education and industry applications. It is released under the liberal Modified BSD open source license, provides a well-documented API in the Python programming language, and is developed by an active, international team of collaborators. In this paper we highlight the advantages of open source to achieve the goals of the scikit-image library, and we showcase several real-world image processing applications that use scikit-image. More information can be found on the project homepage, http://scikit-image.org.

Introduction

In our data-rich world, images represent a significant subset of all measurements made. Examples include DNA microarrays, microscopy slides, astronomical observations, satellite maps, robotic vision capture, synthetic aperture radar images, and higher-dimensional images such as 3-D magnetic resonance or computed tomography imaging. Exploring these rich data sources requires sophisticated software tools that should be easy to use, free of charge and restrictions, and able to address all the challenges posed by such a diverse field of analysis.

This paper describes scikit-image, a collection of image processing algorithms implemented in the Python programming language by an active community of volunteers and available under the liberal BSD Open Source license. The rising popularity of Python as a scientific programming language, together with the increasing availability of a large eco-system of complementary tools, makes it an ideal environment in which to produce an image processing toolkit.

The project aims are:

1. To provide high quality, well-documented and easy-to-use implementations of common image processing algorithms.

Such algorithms are essential building blocks in many areas of scientific research, algorithmic comparisons and data exploration. In the context of reproducible science, it is important to be able to inspect any source code used for algorithmic flaws or mistakes. Additionally, scientific research often requires custom modification of standard algorithms, further emphasizing the importance of open source.

2. To facilitate education in image processing.

The library allows students in image processing to learn algorithms in a hands-on fashion by adjusting parameters and modifying code. In addition, a novice module is provided, not only for teaching programming in the “turtle graphics” paradigm, but also to familiarize users with image concepts such as color and dimensionality. Furthermore, the project takes part in the yearly Google Summer of Code program1, where students learn about image processing and software engineering through contributing to the project.

3. To address industry challenges.

High quality reference implementations of trusted algorithms provide industry with a reliable way of attacking problems without having to expend significant energy in re-implementing algorithms already available in commercial packages. Companies may use the library entirely free of charge, and have the option of contributing changes back, should they so wish.

Getting started

One of the main goals of scikit-image is to make it easy for any user to get started quickly—especially users already familiar with Python’s scientific tools. To that end, the basic image is just a standard NumPy array, which exposes pixel data directly to the user. A new user can simply load an image from disk (or use one of scikit-image’s sample images), process that image with one or more image filters, and quickly display the results:

from skimage import data, io, filter image = data.coins() # or any NumPy array! edges = filter.sobel(image) io.imshow(edges)

The above demonstration loads data.coins, an example image shipped with scikit-image. For a more complete example, we import NumPy for array manipulation and matplotlib for plotting (Van der Walt, Colbert & Varoquaux, 2011; Hunter, 2007). At each step, we add the picture or the plot to a matplotlib figure shown in Fig. 1.

import numpy as np import matplotlib.pyplot as plt # Load a small section of the image. image = data.coins()[0:95, 70:370] fig, axes = plt.subplots(ncols=2, nrows=3, figsize=(8, 4)) ax0, ax1, ax2, ax3, ax4, ax5 = axes.flat ax0.imshow(image, cmap=plt.cm.gray) ax0.set_title('Original', fontsize=24) ax0.axis('off')

Since the image is represented by a NumPy array, we can easily perform operations such as building a histogram of the intensity values.

Figure 1 Illustration of several functions available in scikit-image: adaptive threshold, local maxima, edge detection and labels.

The use of NumPy arrays as our data container also enables the use of NumPy’s built-in histogram function.

# Histogram. values, bins = np.histogram(image, bins=np.arange(256)) ax1.plot(bins[:-1], values, lw=2, c='k') ax1.set_xlim(xmax=256) ax1.set_yticks([0, 400]) ax1.set_aspect(.2) ax1.set_title('Histogram', fontsize=24)

To divide the foreground and background, we threshold the image to produce a binary image. Several threshold algorithms are available. Here, we employ filter.threshold_adaptive where the threshold value is the weighted mean for the local neighborhood of a pixel.

# Apply threshold. from skimage.filter import threshold_adaptive bw = threshold_adaptive(image, 95, offset=-15) ax2.imshow(bw, cmap=plt.cm.gray) ax2.set_title('Adaptive threshold', fontsize=24) ax2.axis('off')

We can easily detect interesting features, such as local maxima and edges. The function feature.peak_local_max can be used to return the coordinates of local maxima in an image.

# Find maxima. from skimage.feature import peak_local_max coordinates = peak_local_max(image, min_distance=20) ax3.imshow(image, cmap=plt.cm.gray) ax3.autoscale(False) ax3.plot(coordinates[:, 1], coordinates[:, 0], c='r.') ax3.set_title('Peak local maxima', fontsize=24) ax3.axis('off')

Next, a Canny filter (filter.canny) (Canny, 1986) detects the edge of each coin.

# Detect edges. from skimage import filter edges = filter.canny(image, sigma=3, low_threshold=10, high_threshold=80) ax4.imshow(edges, cmap=plt.cm.gray) ax4.set_title('Edges', fontsize=24) ax4.axis('off')

Then, we attribute to each coin a label (morphology.label) that can be used to extract a sub-picture. Finally, physical information such as the position, area, eccentricity, perimeter, and moments can be extracted using measure.regionprops.

# Label image regions. from skimage.measure import regionprops import matplotlib.patches as mpatches from skimage.morphology import label label_image = label(edges) ax5.imshow(image, cmap=plt.cm.gray) ax5.set_title('Labeled items', fontsize=24) ax5.axis('off') for region in regionprops(label_image): # Draw rectangle around segmented coins. minr, minc, maxr, maxc = region.bbox rect = mpatches.Rectangle((minc, minr), maxc - minc, maxr - minr, fill=False, edgecolor='red', linewidth=2) ax5.add_patch(rect) plt.tight_layout() plt.show()

scikit-image thus makes it possible to perform sophisticated image processing tasks with only a few function calls.

Library overview

The scikit-image project started in August of 2009 and has received contributions from more than 100 individuals.2 The package can be installed on all major platforms (e.g., BSD, GNU/Linux, OS X, Windows) from, amongst other sources, the Python Package Index (PyPI),3 Continuum Analytics Anaconda,4 Enthought Canopy,5 Python(x,y),6 NeuroDebian (Halchenko & Hanke, 2012) and GNU/Linux distributions such as Ubuntu.7 In March 2014 alone, the package was downloaded more than 5000 times from PyPI.8

As of version 0.10, the package contains the following sub-modules:

• color: Color space conversion.

• data: Test images and example data.

• draw: Drawing primitives (lines, text, etc.) that operate on NumPy arrays.

• exposure: Image intensity adjustment, e.g., histogram equalization, etc.

• feature: Feature detection and extraction, e.g., texture analysis, corners, etc.

• filter: Sharpening, edge finding, rank filters, thresholding, etc.

• graph: Graph-theoretic operations, e.g., shortest paths.

• io: Wraps various libraries for reading, saving, and displaying images and video, such as Pillow9 and FreeImage.10

• measure: Measurement of image properties, e.g., similarity and contours.

• morphology: Morphological operations, e.g., opening or skeletonization.

• novice: Simplified interface for teaching purposes.

• restoration: Restoration algorithms, e.g., deconvolution algorithms, denoising, etc.

• segmentation: Partitioning an image into multiple regions.

• transform: Geometric and other transforms, e.g., rotation or the Radon transform.

• viewer: A simple graphical user interface for visualizing results and exploring parameters.

For further details on each module, we refer readers to the API documentation online.11

Data format and pipelining

scikit-image represents images as NumPy arrays (Van der Walt, Colbert & Varoquaux, 2011), the de facto standard for storage of multi-dimensional data in scientific Python. Each array has a dimensionality, such as 2 for a 2-D grayscale image, 3 for a 2-D multi-channel image, or 4 for a 3-D multi-channel image; a shape, such as (M, N, 3) for an RGB color image with M vertical and N horizontal pixels; and a numeric data type, such as float for continuous-valued pixels and uint8 for 8-bit pixels. Our use of NumPy arrays as the fundamental data structure maximizes compatibility with the rest of the scientific Python ecosystem. Data can be passed as-is to other tools such as NumPy, SciPy, matplotlib, scikit-learn (Pedregosa et al., 2011), Mahotas (Coelho, 2013), OpenCV, and more.

Images of differing data-types can complicate the construction of pipelines. scikit-image follows an “Anything In, Anything Out” approach, whereby all functions are expected to allow input of an arbitrary data-type but, for efficiency, also get to choose their own output format. Data-type ranges are clearly defined. Floating point images are expected to have values between 0 and 1 (unsigned images) or −1 and 1 (signed images), while 8-bit images are expected to have values in {0, 1, 2, …, 255}. We provide utility functions, such as img_as_float, to easily convert between data-types.

Development practices

The purpose of scikit-image is to provide a high-quality library of powerful, diverse image processing tools free of charge and restrictions. These principles are the foundation for the development practices in the scikit-image community.

The library is licensed under the Modified BSD license, which allows unrestricted redistribution for any purpose as long as copyright notices and disclaimers of warranty are maintained (Wilson, 2012). It is compatible with GPL licenses, so users of scikit-image can choose to make their code available under the GPL. However, unlike the GPL, it does not require users to open-source derivative work (BSD is not a so-called copyleft license). Thus, scikit-image can also be used in closed-source, commercial environments.

The development team of scikit-image is an open community that collaborates on the GitHub platform for issue tracking, code review, and release management.12Google Groups is used as a public discussion forum for user support, community development, and announcements.13

scikit-image complies with the PEP8 coding style standard (Van Rossum, Warsaw & Coghlan, 2001) and the NumPy documentation format (Van der Walt & NumPy developers, 2008) in order to provide a consistent, familiar user experience across the library similar to other scientific Python packages. As mentioned earlier, the data representation used is n-dimensional NumPy arrays, which ensures broad interoperability within the scientific Python ecosystem. The majority of the scikit-image API is intentionally designed as a functional interface which allows one to simply apply one function to the output of another. This modular approach also lowers the barrier of entry for new contributors, since one only needs to master a small part of the entire library in order to make an addition.

We ensure high code quality by a thorough review process using the pull request interface on GitHub.14 This enables the core developers and other interested parties to comment on specific lines of proposed code changes, and for the proponents of the changes to update their submission accordingly. Once all the changes have been approved, they can be merged automatically. This process applies not just to outside contributions, but also to the core developers.

The source code is mainly written in Python, although certain performance critical sections are implemented in Cython, an optimising static compiler for Python (Behnel et al., 2011). scikit-image aims to achieve full unit test coverage, which is above 87% as of release 0.10 and continues to rise. A continuous integration system15 automatically checks each commit for unit test coverage and failures on both Python 2 and Python 3. Additionally, the code is analyzed by flake8 (Cordasco, 2010) to ensure compliance with the PEP8 coding style standards (Van Rossum, Warsaw & Coghlan, 2001). Finally, the properties of each public function are documented thoroughly in an API reference guide, embedded as Python docstrings and accessible through the official project homepage or an interactive Python console. Short usage examples are typically included inside the docstrings, and new features are accompanied by longer, self-contained example scripts added to the narrative documentation and compiled to a gallery on the project website. We use Sphinx (Brandl, 2007) to automatically generate both library documentation and the website.

The development master branch is fully functional at all times and can be obtained from GitHub12. The community releases major updates as stable versions approximately every six months. Major releases include new features, while minor releases typically contain only bug fixes. Going forward, users will be notified about API-breaking changes through deprecation warnings for two full major releases before the changes are applied.

Usage examples

Research

Often, a disproportionately large component of research involves dealing with various image data-types, color representations, and file format conversion. scikit-image offers robust tools for converting between image data-types (Microsoft, 1995; Munshi & Leech, 2010; Paeth, 1990) and to do file input/output (I/O) operations. Our purpose is to allow investigators to focus their time on research, instead of expending effort on mundane low-level tasks.

The package includes a number of algorithms with broad applications across image processing research, from computer vision to medical image analysis. We refer the reader to the current API documentation for a full listing of current capabilities16. In this section we illustrate two real-world usage examples of scikit-image in scientific research.

First, we consider the analysis of a large stack of images, each representing drying droplets containing nanoparticles (see Fig. 2). As the drying proceeds, cracks propagate from the edge of the drop to its center. The aim is to understand crack patterns by collecting statistical information about their positions, as well as their time and order of appearance. To improve the speed at which data is processed, each experiment, constituting an image stack, is automatically analysed without human intervention. The contact line is detected by a circular Hough transform (transform.hough_circle) providing the drop radius and its center. Then, a smaller concentric circle is drawn (draw.circle_perimeter) and used as a mask to extract intensity values from the image. Repeating the process on each image in the stack, collected pixels can be assembled to make a space–time diagram. As a result, a complex stack of images is reduced to a single image summarizing the underlying dynamic process.

Figure 2 scikit-image is used to track the propagation of cracks (black lines) in a drying colloidal droplet.

The sequence of pictures shows the temporal evolution of the system with the drop contact line, in green, detected by the Hough transform and the circle, in white, used to extract an annulus of pixel intensities. The result shown illustrates the angular position of cracks and their time of appearance.

Next, in regenerative medicine research, scikit-image is used to monitor the regeneration of spinal cord cells in zebrafish embryos (Fig. 3). This process has important implications for the treatment of spinal cord injuries in humans (Bhatt et al., 2004; Thuret, Moon & Gage, 2006).

To understand how spinal cords regenerate in these animals, injured cords are subjected to different treatments. Neuronal precursor cells (labeled green in Fig. 3A) are normally uniformly distributed across the spinal cord. At the wound site, they have been removed. We wish to monitor the arrival of new cells at the wound site over time. In Fig. 3, we see an embryo two days after wounding, with precursor cells beginning to move back into the wound site (the site of minimum fluorescence). The measure.profile_line function measures the fluorescence along the cord, directly proportional to the number of cells. We can thus monitor the recovery process and determine which treatments prevent or accelerate recovery.

Figure 3 The measure.profile_line function being used to track recovery in spinal cord injuries.

(A) An image of fluorescently-labeled nerve cells in an injured zebrafish embryo. (B) The automatically determined region of interest. The SciPy library was used to determine the region extent (Oliphant, 2007; Jones, Oliphant & Peterson, 2001), and functions from the scikit-image draw module were used to draw it. (C) The image intensity along the line of interest, averaged over the displayed width.

Education

scikit-image’s simple, well-documented application programming interface (API) makes it ideal for educational use, either via self-taught exploration or formal training sessions.

The online gallery of examples not only provides an overview of the functionality available in the package but also introduces many of the algorithms commonly used in image processing. This visual index also helps beginners overcome a common entry barrier: locating the class (denoising, segmentation, etc.) and name of operation desired, without being proficient with image processing jargon. For many functions, the documentation includes links to research papers or Wikipedia pages to further guide the user.

Demonstrating the broad utility of scikit-image in education, thirteen-year-old Rishab Gargeya of the Harker School won the Synopsys Silicon Valley Science and Technology Championship using scikit-image in his project, “A software based approach for automated pathology diagnosis of diabetic retinopathy in the human retina” (science-fair.org, 2014).

We have delivered image processing tutorials using scikit-image at various annual scientific Python conferences, such as PyData 2012, SciPy India 2012, and EuroSciPy 2013. Course materials for some of these sessions are found in Haenel, Gouillart & Varoquaux (2014) and are licensed under the permissive CC-BY license (Creative Commons, 2013). These typically include an introduction to the package and provide intuitive, hands-on introductions to image processing concepts. The well documented application programming interface (API) along with tools that facilitate visualization contribute to the learning experience, and make it easy to investigate the effect of different algorithms and parameters. For example, when investigating denoising, it is easy to observe the difference between applying a median filter (filter.rank.median) and a Gaussian filter (filter.gaussian_filter), demonstrating that a median filter preserves straight lines much better.

Finally, easy access to readable source code gives users an opportunity to learn how algorithms are implemented and gives further insight into some of the intricacies of a fast Python implementation, such as indexing tricks and look-up tables.

Industry

Due to the breadth and maturity of its code base, as well as the its commercial-friendly license, scikit-image is well suited for industrial applications.

BT Imaging (http://www.btimaging.com) designs and builds tools that use photoluminescence (PL) imaging for photovoltaic applications. PL imaging can characterize the quality of multicrystalline silicon wafers by illuminating defects that are not visible under standard viewing conditions. Figure 4A shows an optical image of a silicon wafer, and Fig. 4B shows the same wafer using PL imaging. In Fig. 4C, the wafer defects and impurities have been detected through automated image analysis. scikit-image plays a key role in the image processing pipeline. For example, a Hough transform (transform.hough_line) finds the wafer edges in order to segment the wafer from the background. scikit-image is also used for feature extraction. Crystal defects (dislocations) are detected using a band-pass filter, which is implemented as a Difference of Gaussians (filter.gaussian_filter).

The image processing results are input to machine learning algorithms, which assess intrinsic wafer quality. Solar cell manufacturers can use this information to reject poor quality wafers and thereby increase the fraction of solar cells that have high solar conversion efficiency.

Figure 4 Use of scikit-image to study silicon wafer impurities.

(A) An image of an as-cut silicon wafer before it has been processed into a solar cell. (B) A PL image of the same wafer. Wafer defects, which have a negative impact solar cell efficiency, are visible as dark regions. (C) Image processing results. Defects in the crystal growth (dislocations) are colored blue, while red indicates the presence of impurities.

scikit-image is also applied in a commercial setting for biometric security applications. AICBT Ltd uses multispectral imaging to detect when a person attempts to conceal their identity using a facial mask.17 scikit-image performs file I/O (io.imread), histogram equalization (exposure.equalize_hist), and aligns a visible wavelength image with a thermal image (transform.AffineTransform). The system determines the surface temperature of a subject’s skin and detects situations where the face is being obscured.

Example: image registration and stitching

This section gives a step-by-step outline of how to perform panorama stitching using the primitives found in scikit-image. The full source code is at https://github.com/scikit-image/scikit-image-demos.

Data loading

The “ImageCollection” class provides an easy way of representing multiple images on disk. For efficiency, images are not read until accessed.

from skimage import io ic = io.ImageCollection('data/*')

Figure 5A shows the Petra dataset, which displays the same facade from two different angles. For this demonstration, we will estimate a projective transformation that relates the two images. Since the outer parts of these photographs do not conform well to such a model, we select only the central parts. To further speed up the demonstration, images are downscaled to 25% of their original size.

Figure 5 An example application of scikit-image: image registration and warping to combine overlapping images.

(A) Photographs taken in Petra, Jordan by François Malan. License: CC-BY. (B) Putative matches computed from ORB binary features. (C) Matches filtered using RANSAC. (D) The second input frame (middle) is warped to align with the first input frame (left), yielding the averaged image shown on the right. (E) The final panorama image, registered and warped using scikit-image, blended with Enblend.

from skimage.color import rgb2gray from skimage import transform image0 = rgb2gray(ic[0][:, 500:500+1987, :]) image1 = rgb2gray(ic[1][:, 500:500+1987, :]) image0 = transform.rescale(image0, 0.25) image1 = transform.rescale(image1, 0.25)

Feature detection and matching

“Oriented FAST and rotated BRIEF” (ORB) features (Rublee et al., 2011) are detected in both images. Each feature yields a binary descriptor; those are used to find the putative matches shown in Fig. 5B.

from skimage.feature import ORB, match_descriptors orb = ORB(n_keypoints=1000, fast_threshold=0.05) orb.detect_and_extract(image0) keypoints1 = orb.keypoints descriptors1 = orb.descriptors orb.detect_and_extract(image1) keypoints2 = orb.keypoints descriptors2 = orb.descriptors matches12 = match_descriptors(descriptors1, descriptors2, cross_check=True)

Transform estimation

To filter the matches, we apply RANdom SAmple Consensus (RANSAC) (Fischler & Bolles, 1981), a common method for outlier rejection. This iterative process estimates transformation models based on randomly chosen subsets of matches, finally selecting the model which corresponds best with the majority of matches. The new matches are shown in Fig. 5C.

from skimage.measure import ransac # Select keypoints from the source (image to be # registered) and target (reference image). src = keypoints2[matches12[:, 1]][:, ::-1] dst = keypoints1[matches12[:, 0]][:, ::-1] model_robust, inliers = \ ransac((src, dst), ProjectiveTransform, min_samples=4, residual_threshold=2)

Warping

Next, we produce the panorama itself. The first step is to find the shape of the output image by considering the extents of all warped images.

r, c = image1.shape[:2] # Note that transformations take coordinates in # (x, y) format, not (row, column), in order to be # consistent with most literature. corners = np.array([[0, 0], [0, r], [c, 0], [c, r]]) # Warp the image corners to their new positions. warped_corners = model_robust(corners) # Find the extents of both the reference image and # the warped target image. all_corners = np.vstack((warped_corners, corners)) corner_min = np.min(all_corners, axis=0) corner_max = np.max(all_corners, axis=0) output_shape = (corner_max - corner_min) output_shape = np.ceil(output_shape[::-1])

The images are now warped according to the estimated transformation model. Values outside the input images are set to −1 to distinguish the “background”.

A shift is added to ensure that both images are visible in their entirety. Note that warp takes the inverse mapping as input.

from skimage.color import gray2rgb from skimage.exposure import rescale_intensity from skimage.transform import warp from skimage.transform import SimilarityTransform offset = SimilarityTransform(translation=-corner_min) image0_ = warp(image0, offset.inverse, output_shape=output_shape, cval=-1) image1_ = warp(image1, (model_robust + offset).inverse, output_shape=output_shape, cval=-1)

An alpha channel is added to the warped images before merging them into a single image:

def add_alpha(image, background=-1): """Add an alpha layer to the image. The alpha layer is set to 1 for foreground and 0 for background. """ rgb = gray2rgb(image) alpha = (image != background) return np.dstack((rgb, alpha)) image0_alpha = add_alpha(image0_) image1_alpha = add_alpha(image1_) merged = (image0_alpha + image1_alpha) alpha = merged[..., 3] # The summed alpha layers give us an indication of # how many images were combined to make up each # pixel. Divide by the number of images to get # an average. merged /= np.maximum(alpha, 1)[..., np.newaxis]

The merged image is shown in Fig. 5D. Note that, while the columns are well aligned, the color intensities at the boundaries are not well matched.

Blending

To blend images smoothly we make use of the open source package Enblend (Dersch, 2010), which in turn employs multi-resolution splines and Laplacian pyramids (Burt & Adelson, 1983a; Burt & Adelson, 1983b). The final panorama is shown in Fig. 5E.

Discussion

Related work

In this section, we describe other libraries with similar goals to ours.

Within the scientific Python ecosystem, Mahotas contains many similar functions, and is furthermore also designed to work with NumPy arrays (Coelho, 2013). The major philosophical difference between Mahotas and scikit-image is that Mahotas is almost exclusively written in templated C++, while scikit-image is written in Python and Cython. We feel that our choice lowers the barrier of entry for new contributors. However, thanks to the interoperability between the two provided by the NumPy array data format, users don’t have to choose between them, and can simply use the best components of each.

ImageJ and its batteries-included Fiji distribution are probably the most popular open-source tools for image analysis (Schneider, Rasband & Eliceiri, 2012; Schindelin et al., 2012). Although Fiji’s breadth of functionality is unparalleled, it is centered around interactive, GUI use. For many developers, then, scikit-image offers several advantages. Although Fiji offers a programmable macro mode that supports many scripting languages, many of the macro functions activate GUI elements and cannot run in headless mode. This is problematic for data analysis in high-performance computing cluster environments or web backends, for example. Additionally, Fiji’s inclusive plugin policy results in an inconsistent API and variable documentation quality. Using scikit-image to develop new functionality or to build batch applications for distributed computing is often much simpler, thanks to its consistent API and the wide distribution of the scientific Python stack.

In many respects, the image processing toolbox of the Matlab environment is quite similar to scikit-image. For example, its API is mostly functional and applies to generic multidimensional numeric arrays. However, Matlab’s commercial licensing can be a significant nuisance to users. Additionally, the licensing cost increases dramatically for parallel computing, with per-worker pricing.18 Finally, the closed source nature of the toolbox prevents users from learning from the code or modifying it for specific purposes, which is a common necessity in scientific research. We refer readers back to the Development Practices section for a summary of the practical and philosophical advantages of our open-source licensing.

OpenCV is a BSD-licensed open-source library focused on computer vision, with a separate module for image processing (Bradski, 2000). It is developed in C/C++ and the project’s main aim is to provide implementations for real-time applications. This results in fast implementations with a comparatively high barrier of entry for code study and modification. The library provides interfaces for several high-level programming languages, including Python through the NumPy-array data-type for images. The Python interface is essentially a one-to-one copy of the underlying C/C++ API, and thus image processing pipelines have to follow an imperative programming style. In contrast, scikit-image provides a Pythonic interface with the option to follow an imperative or functional approach. Beyond that, OpenCV’s image processing module is traditionally limited to 2-dimensional imagery.

The choice of image processing package depends on several factors, including speed, code quality and correctness, community support, ecosystem, feature richness, and users’ ability to contribute. Sometimes, advantages in one factor come at the cost of another. For example, our approach of writing code in a high-level language may affect performance, or our strict code review guidelines may hamper the number of features we ultimately provide. We motivate our design decisions for scikit-image in the Development Practices section, and leave readers to decide which library is right for them.

Roadmap

In many open source projects, decisions about future development are made through “rough consensus and working code” (Hoffman, 2014). In scikit-image there are two ways to propose new ideas: through discussion on the mailing list, or as pull requests. The latter route has the advantage of a concrete implementation to guide the conversation, and often mailing list discussions also result in a request for a proof of concept implementation. While conversations are usually led by active developers, the entire community is invited to participate. Once general agreement is reached that the proposed idea aligns with the current project goals and is feasible, work is divided on a volunteer basis. As such, the schedule for completion is often flexible.

The following goals have been identified for the next release of scikit-image:

• Obtain full test coverage.

• Overhaul the functions for image reading/writing.

• Improve the project infrastructure, e.g., create an interactive gallery of examples.

• Add support for graph-based operations.

• Significantly extend higher dimensional (multi-layer) support.

We also invite readers to submit their own feature requests to the mailing list for further discussion.

Conclusion

scikit-image provides easy access to a powerful array of image processing functionality. Over the past few years, it has seen significant growth in both adoption and contribution,19 and the team is excited to collaborate with others to see it grow even further, and to establish it the de facto library for image processing in Python.

We thank Timo Friedrich and Jan Kaslin for providing the zebrafish lesion data. We also acknowledge the efforts of the more than 100 contributors to the scikit-image code base: https://github.com/scikit-image/scikit-image/graphs/contributors.

Additional Information and Declarations

Competing Interests

Author Contributions

1 https://developers.google.com/open-source/soc (accessed 30 March 2014).

2 https://www.ohloh.net/p/scikit-image

3 http://pypi.python.org

4 https://store.continuum.io/cshop/anaconda

5 https://www.enthought.com/products/canopy

6 https://code.google.com/p/pythonxy

7 http://packages.ubuntu.com

8 http://pypi.python.org/pypi/scikit-image (accessed 30 March 2014).

9 http://pillow.readthedocs.org/en/latest/ (accessed 30 May 2015).

10 http://freeimage.sourceforge.net/ (accessed 15 May 2015).

11 http://scikit-image.org/docs/dev/api/api.html

12 https://github.com/scikit-image

13 https://groups.google.com/group/scikit-image

14 https://help.github.com/articles/using-pull-requests (accessed 15 May 2014).

15 https://travis-ci.org, https://coveralls.io (accessed 30 March 2014).

16 http://scikit-image.org/docs/dev (accessed 30 March 2014).

17 http://www.aicbt.com/disguise-detection (accessed 30 March 2014).

18 http://www.mathworks.com.au/products/distriben/description3.html (accessed 9 May 2014).

19 https://www.ohloh.net/p/scikit-image (accessed 15 May 2014).

Neil Yager is an employee of AICBT Ltd; Tony Yu is an employee of Enthought, Inc.

Stéfan van der Walt, Johannes L. Schönberger, Juan Nunez-Iglesias, François Boulogne and Neil Yager conceived and designed the experiments, performed the experiments, wrote the paper, prepared figures and/or tables, reviewed drafts of the paper, wrote software.

Joshua D. Warner, Emmanuelle Gouillart and Tony Yu conceived and designed the experiments, performed the experiments, wrote the paper, reviewed drafts of the paper, wrote software.

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
