# Peer review of "scikit-image: image processing in Python"

_PeerJ, doi:10.7717/peerj.453_

## Round 0.1 · original submission · Minor Revisions

The tool described is a valuable open-source resource available to the biomedical research community. Before we can accept it, please respond to the reviewer concerns.

·

Basic reporting

This paper is an excellent and complete summary of a very important tool for enabling research, education, and industry applications. According to the PeerJ standards, however, submissions should "clearly define the research question…" While this paper clearly states three goals of the scikit-image project, and does an excellent job arguing for their relevance, there is no "research question" in a classical sense. That being said, tools like scikit-image are an increasingly important part of the actual process of doing research, as the authors eloquently point-out.

Experimental design

Again, this paper does not conform to classic notions of "experimental design", but rather gives an excellent summary of a very important software tool and some examples of research that it enables.

Validity of the findings

This paper does not present "findings" in the classical sense of the term, and does not have conclusions based on data or a classical research question. In this sense, it may fit into the category of "case studies", which are specifically mentioned as not fitting into the scope of the journal. I don't think that this aspect of the work can or should be changed: this paper as written is a well-constructed summary of a very important tool for research.

Additional comments

This is an excellent summary of the package, and convinces me that I should invest the time to learn it and use it for some of my own projects! My only comments are minor ones:

- lines 139-140: the GitHub and Google Groups references are a bit confusing. Perhaps these should be footnoted URLs rather than dated references? In any case, I'd move the references to the end of their respective sentences.
- line 146: "guarantees universal interoperability" is a bit strong. I can't think of counterexamples at the moment, but they may exist!
- line 170: the wikipedia reference is confusing.
- line 229: stray closed parenthesis
- line 298: it is very difficult to see the mismatch mentioned here in figure 5d, given how small it is.

I highly recommend this paper for publication after those minor revisions. My above comments in the "Basic Reporting", "Experimental Design", and "Validity of Findings" sections reflect the criteria by which the PeerJ review process asked me to evaluate the paper. It may be that this paper does not fit the editorial scope of PeerJ. If that is the case, there are other journals which may be a better match, and I have confidence that this paper would be quickly accepted.

Reviewer 2 ·

Basic reporting

No comments.

Experimental design

No comments.

Validity of the findings

I do not wish to take away credit from Rishab Gargeya, but I think a smart 13 year-old can handle complex systems quite well and such an achievement does not necessarily demonstrate the shallowness of scikit-image's learning curve.

Additional comments

The authors present scikit-image, a very useful package for image processing in
Python. This package is generally of very high quality and is used by many
already, which demonstrates its usefulness to the community.

MAJOR ISSUES

The main point which I feel is missing from the manuscript are more details on
the Section "Library Contents". This is currently very short.

MINOR ISSUES
A large number of references are just web URLs and not specific documents. For
example, why are references to commercial companies (BT Imaging, Travis CI,...)
then repeated as references when a simple in-text or footnote with the URL
would provide the same content? Some of the others would be improved if the
title of the document was provided in the reference instead of only the URL.
For example, the Microsoft link is to the Direct 3D Data Conversion Rules
document, something that is only apparent after having manually input a complex
URL into a browser (I was working from a printed copy, thus unable to even copy
& paste).

The matplotlib paper should be cited at least once (for the appropriate
citation, see http://matplotlib.org/citing.html).

Page 1/18, line 31: The authors mention "the rising popularity of Python as a
scientific programming language". I wonder whether they have some data for this
claim.

Page 6. lines 85-91: I wonder whether this is the most appropriate location for
this paragraph, it seems awkwardly placed after the "Library Contents" section
title.

Page 6/18, line 92: "currently" should mention a specific version and perhaps
release date to avoid having the paper become outdated and confusing very fast.

Page 7/18, line 146: I would prefer if "universal interoperability" was
rephrased to just "interoperability"

Page 7/18, line 152: I do not think that all the readers will be familiar with
the pull request interface on GH and a short explanation would be in order.

page 8/18, lines 168-173: I found it odd that the Wikipedia "Software Version"
article was referenced here. I do not understand what claim it is backing up.

I also find that the deprecation schedule that the authors describe is very
aggressive for a package that wishes to be foundational. Especially for a user
who does not install from source (for example, when they update their Ubuntu
machine), it will be very easy that they skip a version and do not get a
deprecation warning if they are only there for "approximately every six
months."

page 8/18, line 180: Indeed, file format issues are too often a time sink in
scientific computing. My understanding, however, is that scikit-image only
wraps IO operations from other packages. This fact should be stated explicitly,
providing credit to others and framing the specific advantages of scikit-image
as "providing a common interface".


page 16/18. I am sure that the authors could provide some data for the claim
that their package "has seen significant growth in both adoption and
contribution." I understand the limitations of rough measures (lines of code,
number of commits, &c), but they are informative of trends.

Page 17/18, line 326: There is some character encoding errors for the Halchenko
& Hanke reference.

---

## Round 0.2 · accepted · Accept

Thank you for fully addressing the reviewers' questions and comments.